# Hot alkaline lysis gDNA extraction from formalin-fixed archival tissues

**Erin E. Hahn** [1]*, **Marina Alexander**[1], **Jiri Stiller**[2], **Peter M. Grewe**[3], **Clare E. Holleley**[1]

**1** National Research Collections Australia, Commonwealth Scientific Industrial Research Organisation, Canberra, ACT, Australia, **2** Agriculture and Food, Commonwealth Scientific Industrial Research Organisation, St Lucia, Queensland, Australia, **3** Environment, Commonwealth Scientific Industrial Research Organisation, Hobart, Tasmania, Australia

* erin.hahn@csiro.au

**Data Availability Statement:** All relevant data are within the paper and its Supporting Information files.

**Funding:** Funding for this study was provided by the Environomics CSIRO Future Science Platform

## Abstract

Formalin fixation of natural history specimens and histopathological material has historically been viewed as an impediment to successful genomic analysis. However, the development of extraction methods specifically tailored to contend with heavily crosslinked archival tissues, re-contextualises millions of previously overlooked specimens as viable molecular assets. Here, we present an easy-to-follow protocol for screening archival wet specimens for molecular viability and subsequent genomic DNA extraction suitable for sequencing. The protocol begins with non-destructive assessment of specimen degradation and preservation media conditions to allow both museum curators and researchers to select specimens most likely to yield an acceptable proportion (20–60%) of mappable endogenous DNA during short-read DNA sequencing. The extraction protocol uses hot alkaline lysis in buffer (0.1M NaOH, 1% SDS, pH 13) to simultaneously lyse and de-crosslink the tissue. To maximise DNA recovery, phenol:chloroform extraction is coupled with a small-fragment optimised SPRI bead clean up. Applied to well-preserved archival tissues, the protocol can yield 1–2 μg DNA per 50 mg of tissue with mean fragment sizes typically ranging from 50–150 bp, which is suitable to recover genomic DNA sufficient to reconstruct complete mitochondrial genomes and achieve up to 25X nuclear genome coverage. We provide guidance for read mapping to a reference genome and discuss the limitations of relying on small fragments for SNP genotyping and *de novo* genome assembly. This protocol opens the door to broader-scale genetic and phylogenetic analysis of historical specimens, contributing to a deeper understanding of evolutionary trends and adaptation in response to changing environments.

## Introduction

Museum collections hold a wealth of historical information, functioning as a repository of the planet's biodiversity. Over the last few decades, museum specimens have been increasingly used as sources of genetic material for evolutionary and ecological studies, and have helped to increase our understanding of evolutionary divergence and speciation [1]. Frustratingly, a large proportion of museum specimens have been inaccessible to genomic analyses. These

(grants R-10011 and R-14486) awarded to CEH. Although internally funded by the CSIRO, the funding body did not and will not have a role in study design, data collection and analysis, and decision to publish. The authors sought general comments from members of the funding body in preparation of the manuscript.

**Competing interests:** The authors have declared that no competing interests exist.

specimens, fixed with formalin (3.4% w/v formaldehyde) and typically preserved long-term in ethanol, were long thought to be devoid of sequenceable DNA, thus severely hampering historical genomic research of species primarily preserved in "spirit"–notably fish, reptiles and amphibians. Thanks to recent studies reporting successful sequencing from a diverse range of formalin-preserved specimens [2–8], we now know that formalin preservation does not prohibit DNA sequencing in archival specimens.

A variety of DNA extraction methods have been successfully applied to formalin-preserved museum tissues, including hot alkaline lysis [2, 7, 9], proteinase K digestion with [6] or without [3, 8, 10] guanidine treatment or mediated with a vortex fluidic device [5]. Our group has previously reported that hot alkaline lysis, originally developed for extracting DNA from formalin-fixed and paraffin-embedded tissues [9], yields higher quantities of DNA and proportion of mappable reads compared to proteinase K digestion, especially when applied to poorer quality specimens [7]. In that same paper, we demonstrated a specimen screening method to avoid unnecessary destructive sampling and increase sequencing success. Here we provide an updated and easy-to-follow protocol for selecting specimens, extracting DNA, and determining suitability for advanced sequencing approaches including whole genome sequencing and archival chromatin accessibility sequencing [11]. In this update, we improve DNA recovery by replacing ethanol precipitation of the extracted DNA with a small fragment-optimised SPRI bead purification step. We also provide further guidance around selecting a suitable library preparation method, and maximising read mapping using a trimming-free alignment approach. To showcase this protocol advance, we demonstrate use of the method to yield up to 25X whole genome cover from a set of minute (2–3 mm in length) formalin-fixed Atlantic bluefin tuna larvae (*Thunnus thynnus*) (Fig 1).

## Materials and methods

The protocol described in this peer-reviewed article is published on protocols.io, dx.doi.org/ 10.17504/protocols.io.yxmvm32p6l3p/v1, and is included for printing as S1 File.

### Expected results

Using the protocol as described here, we extracted DNA from 148 (144 fixed in formalin and 4 ethanol-only preserved) *T. thynnus* larvae ranging in length from 2.24–5.34 mm. All larvae were sourced from historical collections from Gulf of Mexico populations held at the National Oceanic and Atmospheric Administration Marine Southeast Fisheries Science Center in Miami, Florida, USA. The formalin-fixed larvae had been collected in 1996 and preserved with 10% seawater buffered formalin and transferred to 95% ethanol after 24 to 48 hours. The ethanol-only preserved larvae had been collected in 2018 and immediately preserved in 95% ethanol.

The total DNA yield from each formalin-fixed larva ranged from 2.88–115.2 ng (mean = 52.77; standard deviation = 24.62), which was a marked improvement over our results using a version of this protocol which utilised ethanol precipitation rather than SPRI beads to concentrate the purified DNA [7] and was similar to the yield from the ethanol-preserved larvae (mean = 52.58 ng). With the 16 formalin-fixed larvae yielding the highest DNA concentrations, as well as the four ethanol-preserved larvae, we prepared libraries for sequencing using the IDT xGen cfDNA and FFPE DNA kit from a uniform 15 ng of input DNA with 13 PCR cycles (Fig 1 and S1 Raw images). We sequenced the resulting libraries each with approximately 125 million paired-end 150 bp reads on the Illumina NovaSeq 6000 S4 platform.

Critically, our whole genome re-sequencing success is supported by using a trimming-free alignment approach as outlined in Hahn et al. [7], which aligns raw reads with the ngskit4b

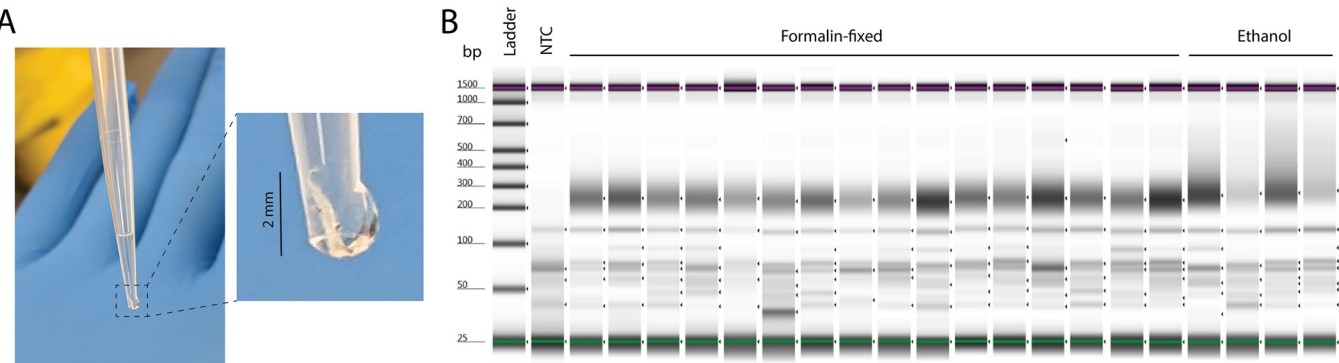

**Fig 1. DNA extracted from formalin-fixed tissue reliably yields high-quality next generation sequencing libraries even from low tissue input.** Using the described method on tuna larvae ranging in size from 2–5 mm (**A**), we extracted DNA and prepared libraries with the IDT xGen cfDNA and FFPE DNA kit. (**B**) Tapestation D1000 traces of uncleaned libraries show pools of library fragments ranging in size from approximately 175–350 bp for formalin-fixed larvae and a more diffuse pool with a lower size boundary of approximately 200 bp for ethanol-preserved larvae.

tool suite version 200218 (https://github.com/kit4b). Specifically here, we aligned to a hard-masked reference genome for *T. albacares* (yellowfin tuna, fThuAlb1.1) using the kalign function with options -c25 (--minchimeric = <int>; minimum chimeric length as a percentage of probe length) -l25 (--minacceptreadlen = <int>; after any end trimming only accept read for further processing if read is at least this length) -d50 (--pairminlen = <int>; accept paired end alignments with observed insert sizes of at least this) -U4 (--pemode = <int>; paired end processing mode: 4 –paired end no orphan recovery treating orphan ends as SE). From these alignments, we remove PCR and optical duplicates from the alignments using the MarkDuplicates function of PICARD enabling REMOVE_DUPLICATES = TRUE.

The mean insert size from the 16 formalin-preserved tuna larvae libraries ranged from 87.8–94.1 bp, which was slightly shorter than that of the ethanol-preserved specimens (mean = 122.5 bp). Pairing this extraction protocol with the IDT xGen cfDNA and FFPE kit, between 59.7% and 67.8% of the raw reads mapped to the reference genome compared to a maximum of 21% using alternative library preparation methods [7] and a mean of 51.0% for the ethanol-preserved larvae. Here, we achieved between 17.8X and 25.4X nuclear genome coverage from the formalin-fixed larvae (Table 1 and Fig 2), which again is a marked improvement over the maximum 3.1X nuclear genome coverage we achieved previously [7]. However, coverage is not directly comparable due to differences in genome size and sequencing effort between the two studies. In comparison, we achieved between 28.8X and 35.9X nuclear genome coverage from the ethanol-preserved larvae. Though the formalin-preserved larvae yielded lower coverage than the ethanol-preserved specimens, the coverage is sufficient for downstream applications. Focusing on the mitochondrial genome coverage, we achieved mean coverage of 1434X with the formalin-preserved larvae and 2215X with the ethanol-preserved larvae, demonstrating that the extraction method can be used to retrieve exceptionally high and even coverage of organellar genomes (Table 1 and Fig 2).

While our method enhances DNA yield and mapping success from formalin-preserved archival tissues, results remain contingent on the preservation quality of the specimen. Our protocol emphasises the importance of evaluating specimens to minimise unnecessary destructive sampling. Most critically, a thorough visual examination of internal organs is crucial to avoid sampling specimens displaying signs of decomposition before fixation. Due to variation in preservation practices, we cannot guarantee specimens identified as suitable will

**Table 1. Exemplary DNA yield and mapping results.**

| Sample_ID | Total DNA yield (ng) | Raw Read Pairs (M) | Mean insert Size (bp) | % mapped | Genome coverage (X) | Mito-Genome coverage (X) |
|---|---|---|---|---|---|---|
| FF1 | 115.2 | 134 | 93.0 | 62.8 | 19.8 | 1784 |
| FF2 | 110.7 | 116 | 94.1 | 66.3 | 18.4 | 1263 |
| FF3 | 107.7 | 154 | 92.8 | 65.2 | 23.6 | 1784 |
| FF4 | 105.3 | 114 | 91.1 | 67.8 | 17.8 | 1219 |
| FF5 | 100.8 | 153 | 89.6 | 63.8 | 22.0 | 1321 |
| FF6 | 99.9 | 136 | 91.9 | 64.6 | 20.4 | 1298 |
| FF7 | 98.7 | 135 | 87.8 | 63.0 | 18.8 | 1406 |
| FF8 | 97.8 | 147 | 91.7 | 65.1 | 22.2 | 1693 |
| FF9 | 97.8 | 143 | 91.7 | 62.8 | 20.8 | 1294 |
| FF10 | 95.4 | 122 | 89.5 | 64.7 | 17.8 | 1093 |
| FF11 | 94.2 | 136 | 94.1 | 59.7 | 19.3 | 1178 |
| FF12 | 93.9 | 160 | 93.8 | 67.0 | 25.4 | 1814 |
| FF13 | 93.9 | 139 | 93.9 | 64.5 | 21.2 | 1536 |
| FF14 | 92.7 | 155 | 93.8 | 64.6 | 23.7 | 1391 |
| FF15 | 91.2 | 134 | 92.0 | 62.0 | 19.3 | 1118 |
| FF16 | 91.2 | 153 | 90.1 | 63.8 | 22.3 | 1760 |
| E1 | 54.3 | 188 | 105.1 | 57.8 | 28.9 | 1990 |
| E2 | 62.1 | 234 | 115.1 | 52.8 | 35.9 | 2470 |
| E3 | 57.6 | 188 | 126.9 | 47.8 | 28.8 | 1981 |
| E4 | 36.3 | 214 | 142.9 | 45.4 | 35.1 | 2420 |

Total DNA yield from hot alkaline lysis extraction of 16 formalin-fixed (FF1 –FF16) and four ethanol-preserved (E1 –E4) tuna larvae as well as mapping results aligning reads to the yellowfin tuna genome using a trimming-free approach with kalign. Sequencing volume as raw read pairs, mean mapped insert size, percentage of mapped reads, mean genomic and mitochondrial cover are presented for all 20 samples.

yield sequenceable DNA. Therefore, we recommend conducting a preliminary extraction on a few representative samples to assess DNA yield; more extensive sampling is recommended if the total yield exceeds 200 ng from 50 mg of tissue. We caution that tissue type can affect results and recommend conducting pilot extractions when exploring alternative tissues. Ideally, studies should report preservation quality metrics alongside sequencing results to enhance comprehension of the relationship between specimen quality and sequencing success. Such an understanding can help to guide further protocol refinement.

DNA extracted from formalin-preserved archival specimens tends to be highly fragmented, with a mean fragment length between 50 and 150 bp. Therefore, we recommend taking expected fragmentation into consideration when designing a sequencing strategy. For example, amplicon-based sequencing approaches may be appropriate only when designed to target fragments in this range. Alternatively, bait-capture sequencing methods can be applied with similar considerations made for the expected fragment sizes. We do not recommend *de novo* genome assembly using these data. Care must be taken in selecting an appropriate reference genome, as mapping success will be affected by both reference quality and the evolutionary distance. Additionally, for phylogenetic analysis, consideration must be given to the pangenomic differences between individuals as they may obscure detection of regions absent in the reference.

Sequencing data for the tuna specimens is being made available on the CSIRO Data Access Portal (DAP) and supporting publications will follow.

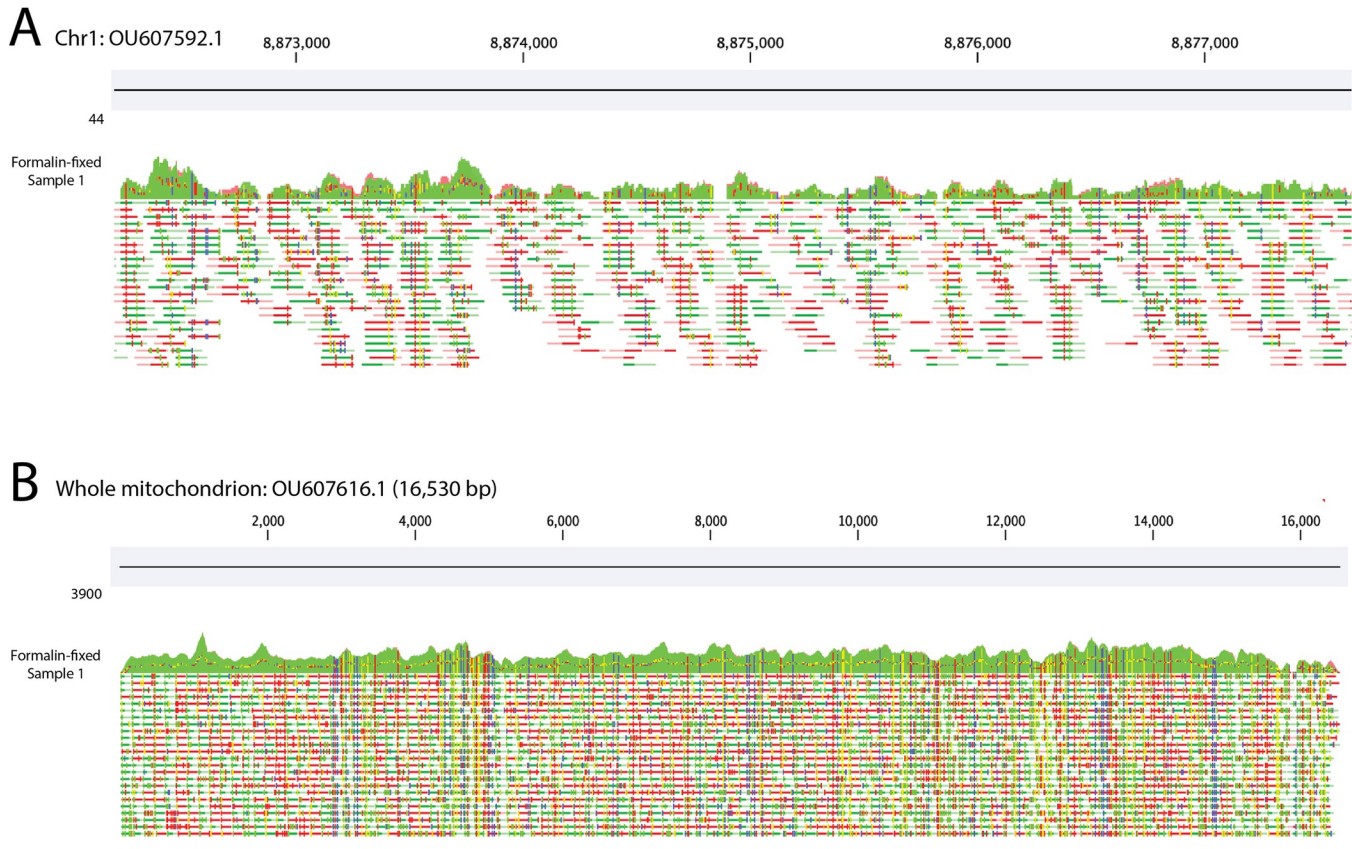

**Fig 2. Alignment of Atlantic bluefin tuna reads mapped to the yellowfin tuna genome using kalign viewed in CLC Genomics Workbench.** Shown are (**A**) a section of the nuclear and (**B**) complete mitochondrial genome for sample 1 from the formalin-fixed larvae. After read deduplication, sample 1 yielded mean 19.8X coverage of the nuclear genome and 1784X coverage of the mitochondrial genome. A coverage graph is shown above in green with individual forward (green) and reverse (red) reads below. Fixed differences between the mapped reads and the reference can be observed in both A & B indicated in blue, yellow and red.

## Supporting information

**S1 File. Step-by-step protocol, also available on protocols.io.**
(PDF)

**S1 Raw images. Raw uncropped images of Tapestation results, in.pdf format.**
(PDF)

## Acknowledgments

We thank Olly Berry and Andrew Young for their leadership within the Environomics Future Science Platform. We also acknowledge the efforts of Glenn Zapfe, Pamela J. Bond and Kristen Walter from National Oceanic and Atmospheric Administration National Marine Fisheries Service in locating and sharing the historical tuna larvae. We thank Sahan Jayasinghe and Anna Kearns for their helpful comments on the manuscript draft. We thank the Australian Genome Research Facility for their conversations around sequencing. We would like to acknowledge the contribution of Bioplatforms Australia in the generation of data used in this publication. Bioplatforms Australia is enabled by NCRIS.

## Author Contributions

**Conceptualization:** Erin E. Hahn, Peter M. Grewe, Clare E. Holleley.

**Data curation:** Erin E. Hahn, Jiri Stiller.

**Formal analysis:** Erin E. Hahn, Jiri Stiller.

**Funding acquisition:** Clare E. Holleley.

**Investigation:** Erin E. Hahn.

**Methodology:** Erin E. Hahn, Marina Alexander.

**Project administration:** Clare E. Holleley.

**Resources:** Peter M. Grewe, Clare E. Holleley.

**Supervision:** Clare E. Holleley.

**Validation:** Erin E. Hahn.

**Writing – original draft:** Erin E. Hahn, Marina Alexander, Peter M. Grewe, Clare E. Holleley.

**Writing – review & editing:** Erin E. Hahn, Marina Alexander, Clare E. Holleley.

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
