## [Decision Letter · Decision Letter 0]

22 Nov 2023

PONE-D-23-29204Hot alkaline lysis gDNA extraction from formalin-fixed archival tissuesPLOS ONE

Dear Dr. Hahn,

Thank you for submitting your manuscript to PLOS ONE. After careful consideration, we feel that it has merit but does not fully meet PLOS ONE’s publication criteria as it currently stands. Therefore, we invite you to submit a revised version of the manuscript that addresses the points raised during the review process.

The proposed protocol is useful and deserves publication after minor changes. Please revise the manuscript considering the reviewer's comments. Moreover, a more in depth discussion about the limitations of the proposed methods and possible further improvements would be appreciated.

We look forward to receiving your revised manuscript.

Kind regards,

Aldo Corriero, Ph.D.

Academic Editor

PLOS ONE

“We thank Olly Berry and Andrew Young for their leadership within the Environomics Future Science Platform. We also acknowledge the efforts of Glenn Zapfe, Pamela J. Bond and Kristen Walter from National Oceanic and Atmospheric Administration National Marine Fisheries Service in locating and sharing the historical tuna larvae. We thank Sahan Jayasinghe and Anna Kearns for their helpful comments on the manuscript draft. We thank the Australian Genome Research Facility for their conversations around sequencing. We would like to acknowledge the contribution of Bioplatforms Australia in the generation of data used in this publication. Bioplatforms Australia is enabled by NCRIS. Funding for this study was provided by the Environomics CSIRO Future Science Platform (grants R-10011 and R-14486) awarded to CEH.”

“Funding for this study was provided by the Environomics CSIRO Future Science Platform (grants R-10011 and R-14486) awarded to CEH. Although internally funded by the CSIRO, the funding body did not and will not have a role in study design, data collection and analysis, and decision to publish. The authors sought general comments from members of the funding body in preparation of the manuscript.”

4. We note that Figures 1A and 1B in your submission contain copyrighted images. All PLOS content is published under the Creative Commons Attribution License (CC BY 4.0), which means that the manuscript, images, and Supporting Information files will be freely available online, and any third party is permitted to access, download, copy, distribute, and use these materials in any way, even commercially, with proper attribution. For more information, see our copyright guidelines: http://journals.plos.org/plosone/s/licenses-and-copyright.

1. You may seek permission from the original copyright holder of Figures 1A and 1B to publish the content specifically under the CC BY 4.0 license.

Reviewers' comments:

Reviewer's Responses to Questions

**Comments to the Author**

1. Does the manuscript report a protocol which is of utility to the research community and adds value to the published literature?

Reviewer #1: Yes

2. Has the protocol been described in sufficient detail?

To answer this question, please click the link to protocols.io in the Materials and Methods section of the manuscript (if a link has been provided) or consult the step-by-step protocol in the Supporting Information files.

The step-by-step protocol should contain sufficient detail for another researcher to be able to reproduce all experiments and analyses.

Reviewer #1: Yes

3. Does the protocol describe a validated method?

Reviewer #1: Yes

4. If the manuscript contains new data, have the authors made this data fully available?

Reviewer #1: Yes

**5. Is the article presented in an intelligible fashion and written in standard English?**

Reviewer #1: Yes

6. Review Comments to the Author

Reviewer #1: The manuscript “Hot alkaline lysis gDNA extraction from formalin-fixed archival tissues” reports about the improvement of genetic analysis of tissues archived in formalin or ethanol. This improvement appears to be an outcome of modifications in DNA extraction from archival specimens, selection of appropriate NGS technology and reference genome. The protocol improvement is substantial and has potential for its publication in the prestigious journal “PLOS ONE”.

Minor comments:

1. In my point of view, it will be better if the authors discuss about the further improvement of the protocol keeping in mind its expected limitations such as decrease in DNA quantity and/or fragment size with every step of purification and its impact on genetic information obtained.

2. Line 58: Kindly remove (Straube) (6)

3. Line 72: Is it necessary to have the word “cover” in “recover up to 25X whole genome cover from”

4. Line 144: Kindly change “reverse (read)” to “reverse (red)”

5. Line 147: Kindly change “Due to the high degree of DNA fragmentation” to “Due to high degree of DNA fragmentation”

6. Line 159 – 162: Is it OK to write down the below sentences for Supporting information as such?

The protocol in PDF format available from protocols.io must be provided as Supporting Information file 1, with the caption: S1: Step-by-step protocol, also available on protocols.io

7. PLOS authors have the option to publish the peer review history of their article (what does this mean?). If published, this will include your full peer review and any attached files.

Reviewer #1: **Yes: **Dr. Muhammad Imran

Associate Professor

Institute of Biochemistry & Biotechnology

University of Veterinary & Animal Sciences - Lahore

Pakistan

---

## [Author Response · Author response to Decision Letter 0]

6 Dec 2023

We have made a few minor changes throughout to adhere to the style requirements, including removing the post codes from the author affiliations.

Please remove any funding-related text from the manuscript and let us know how you would like to update your Funding Statement.

We have removed funding details from the Acknowledgments Section. 

3. PLOS ONE now requires that authors provide the original uncropped and unadjusted images underlying all blot or gel results reported in a submission’s figures or Supporting Information files. 

We have included the requested file as S1_raw_images.pdf.

4. We note that Figures 1A and 1B in your submission contain copyrighted images. We require you to either (1) present written permission from the copyright holder to publish these figures specifically under the CC BY 4.0 license, or (2) remove the figures from your submission.

All images in both figures were created by the authors.

5. Please review your reference list to ensure that it is complete and correct. 

No changes have been made to the reference list.

Response to Reviewers' comments:

Minor comments:

1. In my point of view, it will be better if the authors discuss about the further improvement of the protocol keeping in mind its expected limitations such as decrease in DNA quantity and/or fragment size with every step of purification and its impact on genetic information obtained.

We have added a paragraph and a half addressing these points to the Expected Results section (now lines 149-167).

2. Line 58: Kindly remove (Straube) (6)

Here, we removed “(Straube)” from the text (now line 61).

3. Line 72: Is it necessary to have the word “cover” in “recover up to 25X whole genome cover from”

Changed to “yield” (now line 75).

4. Line 144: Kindly change “reverse (read)” to “reverse (red)”

Amended (now line 150).

5. Line 147: Kindly change “Due to the high degree of DNA fragmentation” to “Due to high degree of DNA fragmentation”

This line was edited to remove this text with the addition of the final paragraph (now line 173).

6. Line 159 – 162: Is it OK to write down the below sentences for Supporting information as such?

The protocol in PDF format available from protocols.io must be provided as Supporting Information file 1, with the caption: S1: Step-by-step protocol, also available on protocols.io

Amended (now line 183).

---

## [Editor Report · Decision Letter 1]

15 Dec 2023

Hot alkaline lysis gDNA extraction from formalin-fixed archival tissues

PONE-D-23-29204R1

Dear Dr. Hahn,

We’re pleased to inform you that your manuscript has been judged scientifically suitable for publication and will be formally accepted for publication once it meets all outstanding technical requirements.

Kind regards,

Aldo Corriero, Ph.D.

Academic Editor

PLOS ONE
---

## [Editor Report · Acceptance letter]

19 Dec 2023

PONE-D-23-29204R1 

PLOS ONE

Dear Dr. Hahn, 

I'm pleased to inform you that your manuscript has been deemed suitable for publication in PLOS ONE. Congratulations! Your manuscript is now being handed over to our production team.

Kind regards, 

on behalf of

Dr. Aldo Corriero 

Academic Editor

PLOS ONE